# A Novel Near-Real-Time GB-InSAR Slope Deformation Monitoring Method

Yuhan Su [1], Honglei Yang [1,*], Junhuan Peng [1], Youfeng Liu [1], Binbin Zhao [2] and Mengyao Shi [1]

1   School of Land Science and Technology, China University of Geosciences, Beijing 100083, China
2   China Electric Power Research Institute Co., Ltd., Beijing 100192, China
*   Correspondence: hongleiyang@cugb.edu.cn

**Abstract:** In the past two decades, ground-based synthetic aperture radars (GB-SARs) have developed rapidly, providing a large amount of SAR data in minutes or even seconds. However, the real-time processing of big data is a challenge for the existing GB-SAR interferometry (GB-InSAR) technology. In this paper, we propose a near-real-time GB-InSAR method for monitoring slope surface deformation. The proposed method uses short baseline SAR data to generate interferograms to improve temporal coherence and reduce atmospheric interference. Then, based on the wrapped phase of each interferogram, a network method is used to estimate and remove systematic errors (such as atmospheric delay, radar center shift error, etc.). After the phase unwrapping, a least squares estimator is used for the overall solution to obtain the initial deformation parameters. When new data are added, a sequential estimator is used to combine the previous processing results and dynamically update the deformation parameters. Sequential estimators could avoid repeated calculations and improve data processing efficiency. Finally, the method is validated with the measured data. The results show that the average deviation between the proposed method and the overall estimation was less than 0.01 mm, which could be considered a consistent estimation accuracy. In addition, the calculation time of the sequential estimator was less sensitive than the total amount of data, and the time-consuming growth rate of each additional period of data was about 1/10 of the overall calculation. In summary, the new method could quickly and effectively obtain high-precision surface deformation information and meet the needs of near-real-time slope deformation monitoring.

**Keywords:** GB-InSAR; near-real-time deformation monitoring; sequential estimation; systematic error correction

## 1. Introduction

Slope deformation is the most direct manifestation of slope instability. Slope deformation monitoring could directly reflect the occurrence, development, and evolution of slope instability. Accordingly, it is of great significance to study slope deformation monitoring technology for slope prediction and slope stability analyses. Commonly used deformation monitoring techniques are based on points, such as leveling, total station, GNSS, etc. These point-based monitoring methods have problems with the inaccurate and difficult layout of monitoring points for high, steep, and complex slopes. Even if intensive monitoring equipment is deployed, it is difficult to obtain the spatial-, continuous-, and whole-process deformation information of a slope. However, this information is very critical for high-risk slopes. Therefore, it is very necessary to use new deformation monitoring technology for slope monitoring.

Ground-based synthetic aperture radar interferometry (GB-InSAR), developed in recent years, is weakly affected by fog, rain, snow, dust, etc., and could achieve all-day, all-weather, and high temporal and spatial resolution observations [1]. It has become a new non-contact monitoring technology for local area deformation. GB-InSAR was first proposed in [2] and used to monitor dam deformation, and then it was widely used in disaster monitoring, such as landslides [3–5], glaciers [6–8], volcanoes [9,10], and structures [11,12].

The existing GB-InSAR time series data processing methods could be divided into two types: post-event and real-time processing. The post-event method is mostly used to monitor slow deformation, and the data processing method draws on the time series processing method of spaceborne InSAR technology, such as permanent scatterer interferometry (PSI) [13,14] and small baseline subsets (SBASs) [15], etc. With the help of high-performance computer equipment, these post-processing methods could realize pseudo-real-time operations on small amounts of data. However, the overall processing of large amounts of data is still a major problem restricting the application of GB-InSAR slope monitoring.

For a very long period of time, real-time GB-InSAR monitoring focused more on real-time data acquisition by equipment [16,17] than real-time data processing. Real-time processing is a new method emerging in recent years, which emphasizes the timeliness of data processing and is mostly used for emergency monitoring, e.g., slope early warning monitoring and emergency rescues [18,19]. In [20], the data grouping processing strategy was adopted, and the adjacent interferograms were directly integrated to realize real-time computation; after accumulating certain data, the interferometric stack was used to re-estimate the deformation to improve the computational accuracy. However, data processing accuracy is proportional to stack size, and it is difficult to consider both accuracy and efficiency. In recent years, sequential estimation methods [21,22] have been proposed to trade off accuracy and efficiency, but the processing methods were based on spaceborne InSAR technology, which is not effective in dealing with errors, such as the atmospheric delay of GB-InSAR. In recent years, a large number of GB-InSAR atmospheric correction methods have been proposed, such as ground control points [23], meteorological data [24], modeling corrections [25], etc. In [26], an atmospheric phase correction method that could be used for near-real-time monitoring was further proposed. However, these methods are all based on phase unwrapping, which inevitably reduces the efficiency and accuracy of data processing.

The main goal of this paper is to propose a new near-real-time processing method for GB-InSAR, which could quickly remove systematic errors and realize dynamic estimation of deformation parameters from real-time acquired SAR data. Based on the spatial distribution characteristics of the GB-InSAR phase, the method uses low-order polynomials to model the systematic error and uses the network constructed by coherent points to estimate the model parameters without phase unwrapping. Then, a sequential estimator is used to calculate the deformation parameters of the newly added data, which could maintain the same accuracy as the overall solution and greatly improve computational efficiency. Finally, the method is validated with the data collected from the pumped-storage power station under construction in Zhen'an County, Shangluo City, Shaanxi Province, China.

## 2. Near-Real-Time GB-InSAR Deformation Measurement Method

The near-real-time GB-InSAR deformation measurement method could be divided into two major steps: the initial dataset deformation estimation and the additional dataset deformation dynamic update. Since the overall solution strategy was adopted first, the amount of initial SAR data $N_1$ was relatively large (not less than 10 scenes) to ensure accuracy. On the contrary, the accuracy of the sequential strategy was independent of the size of the dataset. Therefore, the amount of newly added SAR data $N_2$ was as small as possible, such as $N_2 = 1$, to achieve optimal timeliness. The additional dataset contained some SAR data from the initial dataset, and the number depended on the combination of the interferometric pairs.

The specific algorithm flow is shown in Figure 1. The preprocess included a series of conventional D-InSAR operations, such as registration, interferometry, difference, filtering, etc. Some GB-InSAR devices, especially rotating scanning devices, have deviations in their imaging, so registration is required first. After that, a redundant network of interference pairs was constructed according to the length of the baseline. Finally, adaptive spatial filtering was performed on all interferograms to improve the coherence.

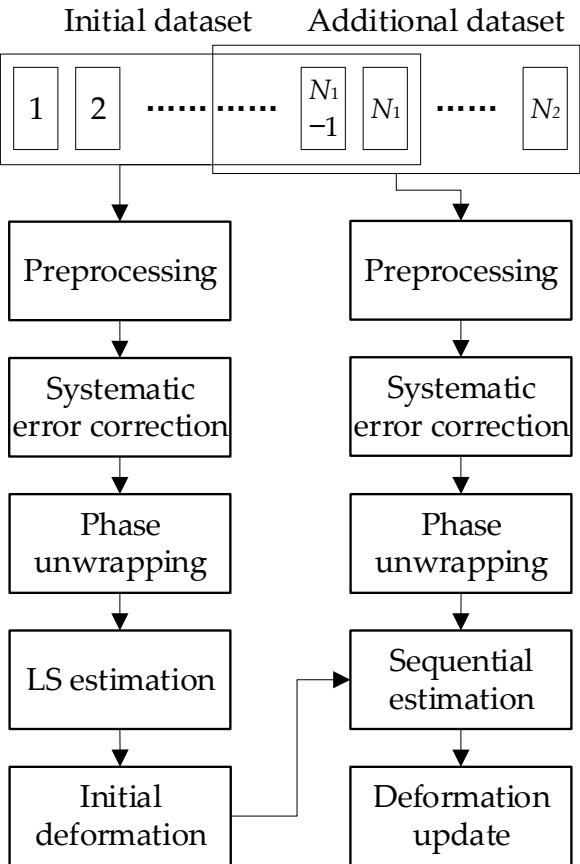

**Figure 1.** Flowchart of the near-real-time GB-InSAR deformation measurement method.

After the preprocessing, the conventional method performs the phase unwrapping first and then removes the systematic errors [15]. On the contrary, we first corrected the systematic error. In this paper, systematic errors refer to modelable errors, including the stratified atmosphere, radar phase center offset errors, etc. It should be noted that if the observation time interval is short enough, the surface deformation does not exceed the quarter wavelength of the radar. When the systematic error phase is corrected, there is no ambiguity in the residual phase (the deformation and random noise); hence phase unwrapping is not necessary for this circumstance. In addition, even if the time interval is long, the gradient of the wrapped phase after the systematic error correction would be reduced, which would improve the accuracy and efficiency of the subsequent unwrapping operation. Finally, the least squares (LS) method was used to obtain the deformation parameters of the initial dataset. In the additional dataset, it was only necessary to process the new data and then use the sequential estimator to combine the results of the initial dataset and update the deformation parameters. The systematic error correction method for the wrapped phase, the LS overall estimation, and the sequential estimation is introduced later.

### 2.1. Systematic Error Correction Method for the Wrapped Phase

In this paper, we provide a systematic error correction framework rather than an exact error model. Specifically, according to the performance characteristics of the systematic errors, different error models could be substituted into our proposed framework to estimate the model parameters and correct the systematic errors. Compared with conventional error model correction methods, the proposed framework is based on the wrapped phase and avoids unwrapping operation errors. In this subsection, we introduce several commonly used systematic error models and the parameter estimation methods of the proposed framework.

The atmospheric delay is the most important error source in GB-InSAR technology and could be divided into the systematic layered atmosphere and random local turbulence [27]. The layered atmosphere is related to the spatial location information of the observed target, such as line-of-sight (LOS) distance [28,29], elevation [25], etc. Therefore, this systematic atmospheric delay phase could be expressed as a multivariate function model:

$$\varphi_{sys}(r, \theta, h, \ldots) = f(\beta_0, \beta_1, \beta_2, \ldots) \tag{1}$$

where $\varphi_{sys}$ is the unwrapped error phase. $(r, \theta, h, \ldots)$ represents the known radar observation parameters, such as the LOS distance from the target to the radar center, the azimuth angle, the relative elevation, etc. $\beta$ is the unknown parameter whose number is equal to the number of terms of the polynomial model. One of the most commonly used models is proposed in [25]:

$$\varphi_{sys} = \beta_0 + \beta_1 r + \beta_2 rh \tag{2}$$

In addition, the slight displacement of the equipment during the observation process would also cause similar systematic errors. In [30], the error phase caused by the radar center offset and the atmospheric delay was comprehensively considered, and the error phase model was described as follows:

$$\varphi_{sys} = \beta_0 + \beta_1 r + \beta_2 r^2 + \beta_3 \frac{h}{r} + \beta_4 \cos\theta - \beta_5 \sin\theta \tag{3}$$

where $\theta$ is the azimuth angle from the radar phase center to the target.

In the above model, the interferometric phase is unwrapped first. Then regression analysis is used to determine the best estimates of the unknown parameters. Finally, Equation (1) is used to calculate the systematic error phase. However, phase unwrapping is inefficient and prone to introducing errors. At this point, the unwrapped systematic error phase could be expressed as follows:

$$\Phi_{sys} = \varphi_{sys} + 2k\pi = W\{f(\beta_0, \beta_1, \beta_2, \ldots)\} \tag{4}$$

where $\Phi_{sys}$ is the unwrapped systematic error phase, $k$ is an unknown integer, and $W\{\cdot\}$ represents a wrapping operator.

The spatial variation of the systematic error phase is relatively stable. If the two points are close enough to have an equal integer number $k$, a new error phase model could be constructed by using the difference between the wrapped phases of the two points. To ensure computational efficiency and accuracy, we simply used a coherence threshold to filter out the high-coherence points and then used Delaunay triangulation to connect these points.

Assuming that an interferogram contains $P$ high correlation points, the constructed triangular network includes $G$ edges, and the number of terms of the polynomial (1) (excluding constant term) is $U$. Then the wrapped phase difference between two points on the $k$th ($k \in [1, G]$) edge can be expressed as:

$$\Delta\Phi_k(dr_k, d\theta_k, dh_k, \ldots) = \Delta f_k(\beta_1, \beta_2, \ldots, \beta_U) \tag{5}$$

d· represents the difference operation, and the constant term $\beta_0$ is eliminated. Let $\underset{G\times 1}{\Delta\Phi} = \begin{bmatrix} \Delta\Phi_1 & \Delta\Phi_2 & \ldots & \Delta\Phi_G \end{bmatrix}^T$, $\underset{G\times U}{\mathbf{D}} = \begin{bmatrix} dr & d\theta & dh & \ldots \end{bmatrix}$ and $\underset{U\times 1}{B} = \begin{bmatrix} \beta_1 & \beta_2 & \ldots & \beta_U \end{bmatrix}^T$, then the differential phase of all the edges can be unified into a matrix $\Delta\Phi$:

$$\underset{G\times 1}{\Delta\Phi} = \underset{G\times U}{\mathbf{D}} \underset{U\times 1}{\mathbf{B}} + \underset{G\times 1}{e}, \underset{G\times G}{\mathbf{P}} \tag{6}$$

where $e$ is the error term, including the random errors and unmodeled interference phase values. $\mathbf{P}$ is the weight matrix of the observations and could be considered an identity matrix in the first calculation. Equation (6) could be solved according to the least squares principle:

$$
\begin{aligned}
\hat{\mathbf{B}} &= \left(\mathbf{D}^{\mathrm{T}}\mathbf{P}\mathbf{D}\right)^{-1}\mathbf{D}^{\mathrm{T}}\mathbf{P}\Delta\boldsymbol{\Phi} \\
\mathbf{Q}_{\hat{\mathbf{B}}} &= \left(\mathbf{D}^{\mathrm{T}}\mathbf{P}\mathbf{D}\right)^{-1} \\
e &= \Delta\boldsymbol{\Phi} - \mathbf{D}\left(\mathbf{D}^{\mathrm{T}}\mathbf{P}\mathbf{D}\right)^{-1}\mathbf{D}^{T}\mathbf{P}\Delta\boldsymbol{\Phi}
\end{aligned}
\tag{7}
$$

where $\hat{\mathbf{B}}$ is the estimate value of the unknown parameter, and $\mathbf{Q}_{\hat{B}}$ is the co-factor matrix.

In the above calculation, the influence of deformation is not considered, and some differential phases have phase ambiguity. In these cases, the $k$ values of the adjacent points are inconsistent, and the interference phase difference deviates abnormally from the model. Therefore, a residual threshold needs to be set to detect and remove these abnormal edges. In addition, the residual error $e$ could be used to determine the weight matrix to weaken the influence of the abnormal edges. In this case, the new weight matrix is $\mathbf{P}_{new} = \mathrm{diag}\left(e_1{}^{-2}, e_2{}^{-2}, \ldots, e_G{}^{-2}\right)$. Then, the observation equation according to Equations (6) and (7) is constructed, and the exact estimated value of the unknown parameter is calculated. By bringing the unknown parameter estimates $\hat{\mathbf{B}}$ into Equation (4), you could further solve the constant term $\hat{\beta}_0$:

$$
\hat{\beta}_0 = \frac{\mathbf{W}\left\{\sum\limits_{k=1}^{G}\left(\Phi_k - f_k\left(\hat{\beta}_1, \hat{\beta}_2, \ldots, \hat{\beta}_U\right)\right)\right\}}{G}
\tag{8}
$$

Finally, after all the unknown parameters are obtained, they are substituted into Equation (4) to estimate the wrapped systematic error phase:

$$
\hat{\Phi}_{sys} = \mathrm{W}\left\{f\left(\hat{\beta}_0, \hat{\beta}_1, \hat{\beta}_2, \ldots, \hat{\beta}_U\right)\right\}.
\tag{9}
$$

*2.2. GB-InSAR Least Squares Overall Estimation*

In order to reduce the decoherence caused by the long baseline, the initial dataset includes $N_1$ ground-based SAR images that are constructed according to the principle of a short baseline to generate $M_1$ scene interferograms. Obviously, $M_1$ satisfies the condition:

$$
(N_1 - 1) \le M_1 \le \frac{N_1(N_1 - 1)}{2}
\tag{10}
$$

If an interferometric pair is composed of $i$ and $j$th scene images, its unwrapped deformation phase could be expressed as $\phi_{i,j}$, where $i, j \in [1, N_1]$ and $i < j$. Then the deformation phase set $\mathbf{L}_1$ (the observation vector) could be expressed as follows:

$$
\mathbf{L}_1 = \begin{bmatrix} \phi_{1,2} & \phi_{1,3} & \phi_{2,3} & \cdots & \phi_{i,j} \end{bmatrix}^{\mathrm{T}}
\tag{11}
$$

In order to determine the weight of observations vector $\mathbf{L}_1$, a conventional approach is to estimate the correlation coefficient in a small window [31]. Generally, the weight of a single observation value could be approximately expressed as follows:

$$
p = \left(\sum_{i=1}^{n}(\phi_i - \overline{\phi})^2/(n-1)\right)^{-1}
\tag{12}
$$

where $\phi_i$ and $\overline{\phi}$ represent the phase of the $i$th point and the average phase of all points in the window, respectively. Moreover, $n$ is the number of points in the window. The weight matrix $\mathbf{P}_1$ of the observation value could be calculated by pixel. However, pixel-by-pixel calculations are inefficient, so it is feasible to simply use an identity matrix as the weight matrix after filtering out the highly coherent points.

Correspondingly, the cumulative deformation phase at the $j$th scene is represented by the unknown $x_{1,j}$ ($j \in [2, N_1]$). The unknown parameter $\mathbf{X}_1$ vector could be expressed as follows:

$$\mathbf{X}_1 = \begin{bmatrix} x_{1,2} & x_{1,2} & x_{1,3} & \cdots & x_{1,N_1} \end{bmatrix}^{\mathrm{T}} \tag{13}$$

If the weight matrix of the observation vector $\mathbf{L}_1$ is $\mathbf{P}_1$ and the error vector is $e_1$, then the observation equation is:

$$\underset{M_1 \times 1}{\mathbf{L}_1} = \underset{M_1 \times (N_1-1)}{\mathbf{C}_1} \underset{(N_1-1) \times 1}{\mathbf{X}_1} + \underset{M_1 \times 1}{e_1} , \mathbf{P}_1 \tag{14}$$

where $\mathbf{C}_1$ is a design matrix of size $M_1 \times (N_1 - 1)$, which is constructed according to the serial numbers of the master and slave images of each interferometric pair. Additionally, $-1$ is set for the master image, and $+1$ is set for the slave image. The general form of $\mathbf{C}_1$ is:

$$\mathbf{C}_1 = \begin{bmatrix} -1 & +1 & 0 & \cdots & 0 \\ -1 & 0 & +1 & \cdots & 0 \\ 0 & -1 & +1 & \cdots & 0 \\ \cdots & \cdots & \cdots & \cdots & \cdots \end{bmatrix} . \tag{15}$$

According to the principle of least squares, the solution of Equation (14) is:

$$\begin{aligned} \hat{\mathbf{X}}_1 &= \left( \mathbf{C}_1^{\mathrm{T}} \mathbf{P}_1 \mathbf{C}_1 \right)^{-1} \mathbf{C}_1^{\mathrm{T}} \mathbf{P}_1 \mathbf{L}_1 \\ \mathbf{Q}_{\hat{\mathbf{X}}_1} &= \left( \mathbf{C}_1^{\mathrm{T}} \mathbf{P}_1 \mathbf{C}_1 \right)^{-1} \end{aligned} \tag{16}$$

where $\hat{\mathbf{X}}_1$ is the estimated value of the cumulative deformation series and $\mathbf{Q}_{\hat{\mathbf{X}}_1}$ is the co-factor matrix. At this point, we obtained the least squares solution for the deformation parameters of the initial SAR dataset.

### 2.3. GB-InSAR Sequential Estimation Method

After obtaining the overall solution of the deformation sequence of the initial SAR dataset (including the $N_1$ scene data), we obtained the newly acquired $N_2$ scene SAR data. Following the same short baseline principle, the newly added SAR constituted interferometric pairs. The unwrapped phase of the initial and new data constituted the observation vectors $\mathbf{L}_1$, $\mathbf{L}_2$, respectively. The cumulative deformation sequences (unknown parameter vectors) were:

$$\begin{aligned} \mathbf{X}_1 &= \begin{bmatrix} x_{1,2} & x_{1,3} & \cdots & x_{1,N_1} \end{bmatrix}^{\mathrm{T}} \\ \mathbf{X}_2 &= \begin{bmatrix} x_{1,N_1+1} & x_{1,N_1+2} & \cdots & x_{1,N_1+N_2} \end{bmatrix}^{\mathrm{T}} \end{aligned} \tag{17}$$

The observation equation according to the conventional ground-based SAR overall solution method was:

$$\begin{aligned} \mathbf{L}_1 &= \mathbf{C}_1 \mathbf{X}_1 + e_1, \mathbf{P}_1 \\ \mathbf{L}_2 &= \mathbf{D}_1 \mathbf{X}_1 + \mathbf{D}_2 \mathbf{X}_2 + e_2, \mathbf{P}_2 \end{aligned} \tag{18}$$

where $\mathbf{C}_1$, $\mathbf{D}_1$ and $\mathbf{D}_2$ are the design matrices. $\mathbf{P}_1$ and $\mathbf{P}_2$ are the weight matrices of the observed values of the two datasets, respectively.

In order to avoid the repeated operation of the overall solution of the conventional ground-based SAR data, we used the sequential estimation strategy to acquire the deformation phase sequence. According to Equation (18), the estimated value $\hat{\mathbf{X}}_1$ and the co-factor matrix $\mathbf{Q}_{\hat{\mathbf{X}}_1}$ of the initial data set were obtained. Substituting $\hat{\mathbf{X}}_1$ as prior information into the observation equation, the new observation equation was:

$$\begin{bmatrix} \hat{\mathbf{X}}_1 \\ \mathbf{L}_2 \end{bmatrix} = \begin{bmatrix} \mathbf{I} & 0 \\ \mathbf{D}_1 & \mathbf{D}_2 \end{bmatrix} \begin{bmatrix} \mathbf{X}'_1 \\ \mathbf{X}_2 \end{bmatrix} + \begin{bmatrix} e_1 \\ e_2 \end{bmatrix} \tag{19}$$

where **I** is the identity matrix; $\mathbf{X}'_1$ and $\mathbf{X}_2$ are the new unknown parameter vectors to be determined. At this point, the weight matrix of the observations was:

$$\mathbf{P}_{\hat{X}_1 L_2} = \begin{bmatrix} \mathbf{Q}_{\hat{X}_1}^{-1} & 0 \\ 0 & \mathbf{P}_2 \end{bmatrix} = \begin{bmatrix} \mathbf{C}_1^{\mathrm{T}} \mathbf{P}_1 \mathbf{C}_1 & 0 \\ 0 & \mathbf{P}_2 \end{bmatrix}. \tag{20}$$

According to the principle of least squares, the estimated value of the unknown parameter could be obtained:

$$\begin{bmatrix} \hat{\mathbf{X}}'_1 \\ \hat{\mathbf{X}}_2 \end{bmatrix} = \begin{bmatrix} \mathbf{C}_1^{\mathrm{T}} \mathbf{P}_1 \mathbf{C}_1 + \mathbf{D}_1^{\mathrm{T}} \mathbf{P}_2 \mathbf{D}_1 & \mathbf{D}_1^{\mathrm{T}} \mathbf{P}_2 \mathbf{D}_2 \\ \mathbf{D}_2^{\mathrm{T}} \mathbf{P}_2 \mathbf{D}_1 & \mathbf{D}_2^{\mathrm{T}} \mathbf{P}_2 \mathbf{D}_2 \end{bmatrix}^{-1} \begin{bmatrix} \mathbf{C}_1^{\mathrm{T}} \mathbf{P}_1 \mathbf{C}_1 \mathbf{X}'_1 + \mathbf{D}_1^{\mathrm{T}} \mathbf{P}_2 \mathbf{L}_2 \\ \mathbf{D}_2^{\mathrm{T}} \mathbf{P}_2 \mathbf{L}_2 \end{bmatrix}. \tag{21}$$

The co-factor matrix for the estimated value was:

$$\mathbf{Q}_{\hat{X}'_1 \hat{X}_2} = \begin{bmatrix} \mathbf{C}_1^{\mathrm{T}} \mathbf{P}_1 \mathbf{C}_1 + \mathbf{D}_1^{\mathrm{T}} \mathbf{P}_2 \mathbf{D}_1 & \mathbf{D}_1^{\mathrm{T}} \mathbf{P}_2 \mathbf{D}_2 \\ \mathbf{D}_2^{\mathrm{T}} \mathbf{P}_2 \mathbf{D}_1 & \mathbf{D}_2^{\mathrm{T}} \mathbf{P}_2 \mathbf{D}_2 \end{bmatrix}^{-1}. \tag{22}$$

## 3. Results

The experimental area was located downstream of the pumped-storage power station under construction (Figure 2a) in Zhen'an County, Shangqiu City, Shaanxi Province. During the experiment, the site underwent blasting excavation (Figure 2b), and the newly excavated slope was exposed (Figure 2c). In addition, the continuous rainfall further aggravated the risk of slope instability. We established a long-term GB-InSAR observation point on a pre-embedded cement pile approximately 50 m from the slope (Figure 2d). The GB-InSAR system used GPRI-II equipment developed by GAMMA Remote Sensing AG in Switzerland. The main technical parameters of GPRI-II are shown in Table 1.

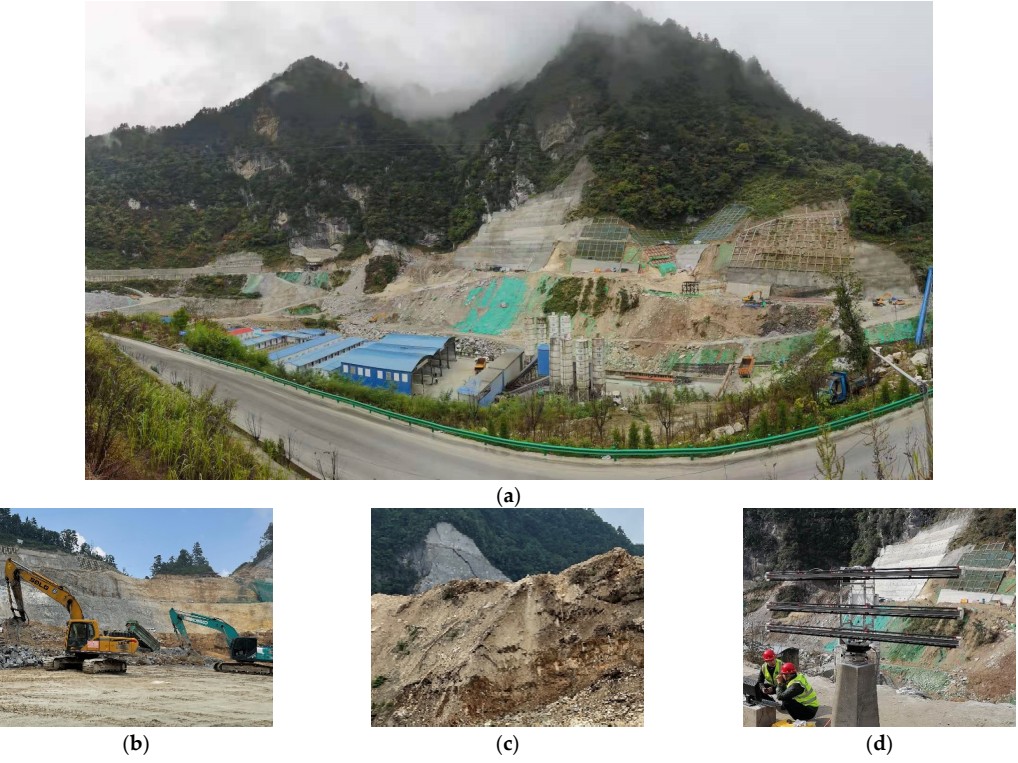

**Figure 2.** (**a**) Panorama of the downstream Zhen'an pumped-storage power station. (**b**) Excavation construction. (**c**) Exposed slope (**d**) GB-InSAR long-term observation point.

**Table 1.** The main parameters of GPRI-II.

| Parameter | Value |
|---|---|
| Center frequency | 17.2 GHz |
| Bandwidth | 2000 MHz |
| Effective measuring range | 50 m to 10 km |
| Range resolution | 0.95 m |
| Azimuth resolution | 0.385 deg |

From 3 April 2021, we conducted a 3-day slope monitoring experiment. The radar equipment performed imaging every 5 min, with an imaging range of 50 m to 425 m, covering the entire exposed slope. A total of 121 single-look complex images (SLCs) were acquired in 3 days (Table 2).

**Table 2.** Observations from 3 April to 5 April 2021.

| NO. | Starting Time | End Time | Number of Images |
|---|---|---|---|
| 1 | 3 April 2021 14:32 | 3 April 2021 16:12 | 21 |
| 2 | 4 April 2021 08:31 | 4 April 2021 15:53 | 44 |
| 3 | 5 April 2021 08:25 | 5 April 2021 15:16 | 56 |

Figure 3a is the intensity image of the acquired SLC data of the first scene, and Figure 3b,c are the interferometric phase map and coherence map generated after the interferometric SLC data of the first two scenes. In the bare soil area, the intensity value was larger, the interference phase distribution was regular, and the coherence was higher. While the intensity value in the dense vegetation area was smaller, the interference phase was randomly distributed, thus, showed complete decoherence.

A total of 357 interferometric pairs were constructed by combining the SLC data of the three adjacent scenes in two pairs. There was a significant phase error in some of the interferograms due to the rapidly changing moisture conditions and the radar center shift caused by the re-installation of the instrument. Therefore, we adopted the method proposed in this paper, the correction model in Equation (3), a correlation threshold of 0.95, and the systematic errors were estimated and removed. The correction effect of five typical areas are seen in Figures 4 and 5, and Table 3. After the systematic error correction, the trend of the wrapped phase was significantly reduced. The average value of the corrected phase was less than 0.05 rad (the corresponding deformation value was less than 0.07 mm), which could meet the accuracy requirements of millimeter-level monitoring. Even for the interference pair (e) with obvious phase jumps, this method had a superior correction result.

**Table 3.** Statistics of the phase before and after the systematic error correction of five typical interferometric pairs.

| | Mean | | Standard Deviations | |
|---|---|---|---|---|
| | Original | Corrected | Original | Corrected |
| (a) | 0.4970 | 0.0038 | 0.4550 | 0.3513 |
| (b) | 0.5259 | 0.0341 | 0.5702 | 0.5182 |
| (c) | −0.6595 | 0.0406 | 0.4578 | 0.3874 |
| (d) | −0.2627 | −0.0305 | 2.0912 | 0.3520 |
| (e) | −0.6186 | 0.0238 | 0.4766 | 0.4401 |

Because the observation time was not continuous, the minimum cost flow (MCF) [32] was used in the experiment for phase unwrapping. To verify the feasibility of not doing an unwrapping, we compared the difference between the wrapped and unwrapped phases (Figure 6). If the difference of a point exceeds 0 (due to the rounding error, the threshold may be slightly greater than 0), the point is considered to have phase ambiguity. In the spatial domain, the points with a large proportion of phase ambiguity were mainly distributed

at the edge or in an isolated area (Figure 6), which is mainly caused by errors introduced with path-dependent phase unwrapping. In the temporal domain, the interferograms composed of different days had a high proportion of ambiguity points (Figure 6b, cross mark), especially between 4 April and 5 April. In addition, the proportion of some interference pairs was slightly higher because of the large deformation on 5 April.

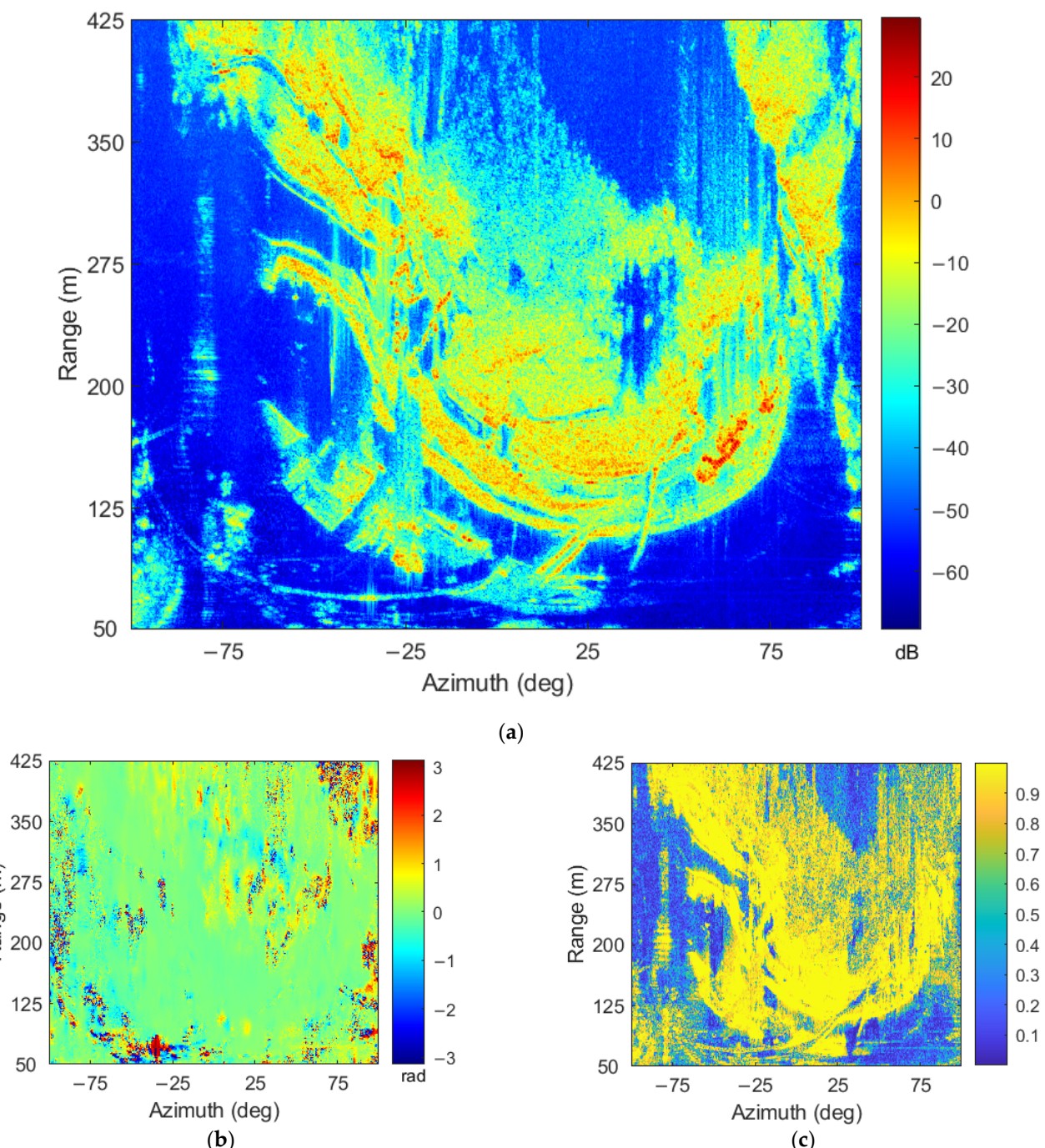

**Figure 3.** (**a**) Intensity image at 14:32 on 3 April 2021. (**b**) Interferogram from 14:32 to 14:37 on 3 April 2021. (**c**) Coherence coefficient image from 14:32 to 14:37 on 3 April 2021.

The 21 SAR data acquired on the first day (3 April 2021) were used as the initial data set, and the coherence threshold was 0.8 to filter out the high coherence points. The conventional SBAS-InSAR (LS overall estimation) was used to process the data to obtain the initial cumulative deformation sequence and average deformation rate. Figure 7 shows

the deformation rate of the experimental area in mm/day, and a positive value indicates that the line-of-sight distance from the point to the radar phase center is decreasing and vice versa. There were some isolated regions with abnormal velocity in the figure, mainly due to phase unwrapping errors. From the figure, an obvious deformation region (the box-selected region in Figure 7) is observed, and its central deformation rate was approximately 11 mm/day.

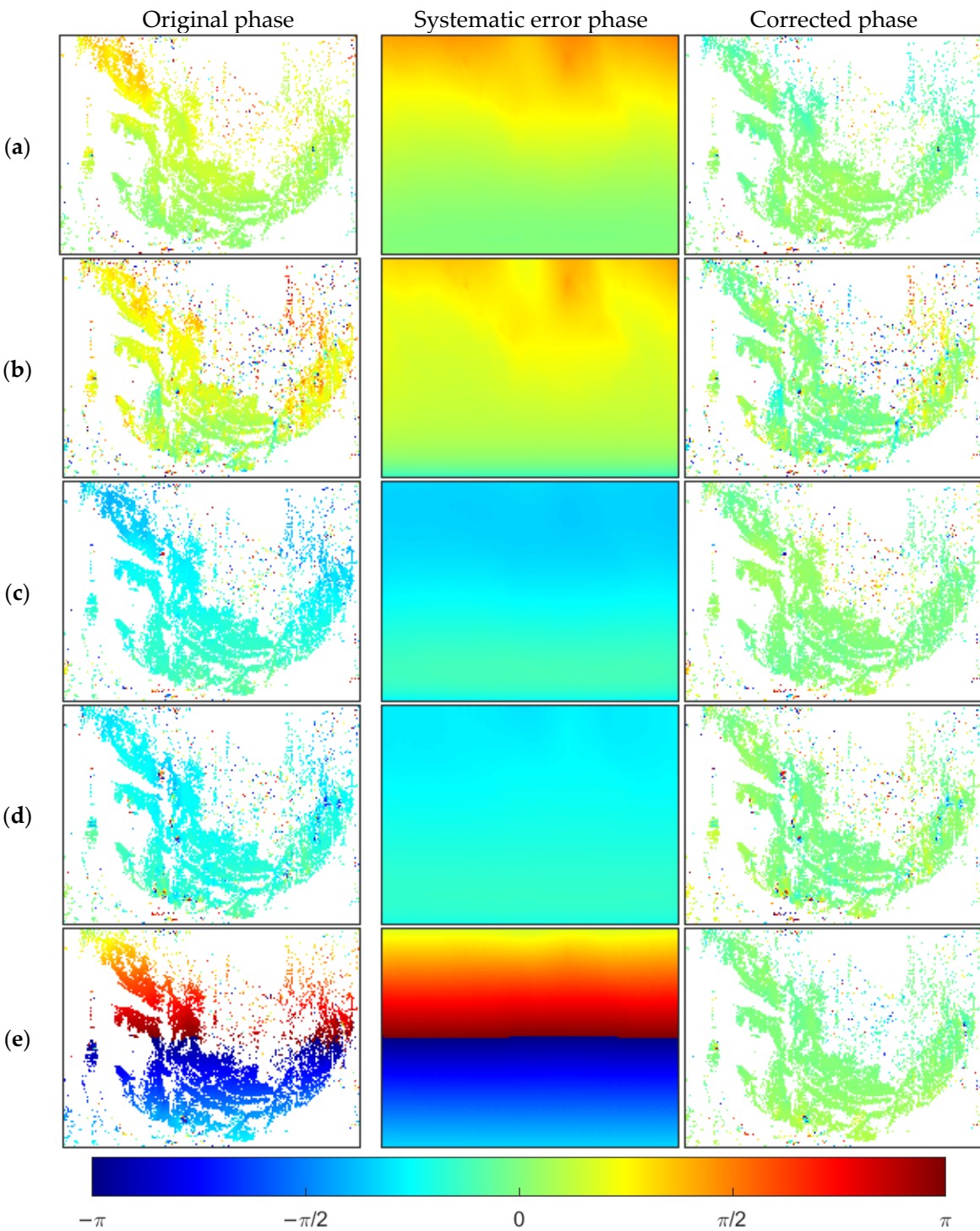

**Figure 4.** Original phase, systematic error phase, and corrected phase of five typical interference pairs in the experimental area. (**a**,**b**) Error related to elevation; (**c**,**d**) overall phase offset; (**e**) radar center deviation caused by blasting construction.

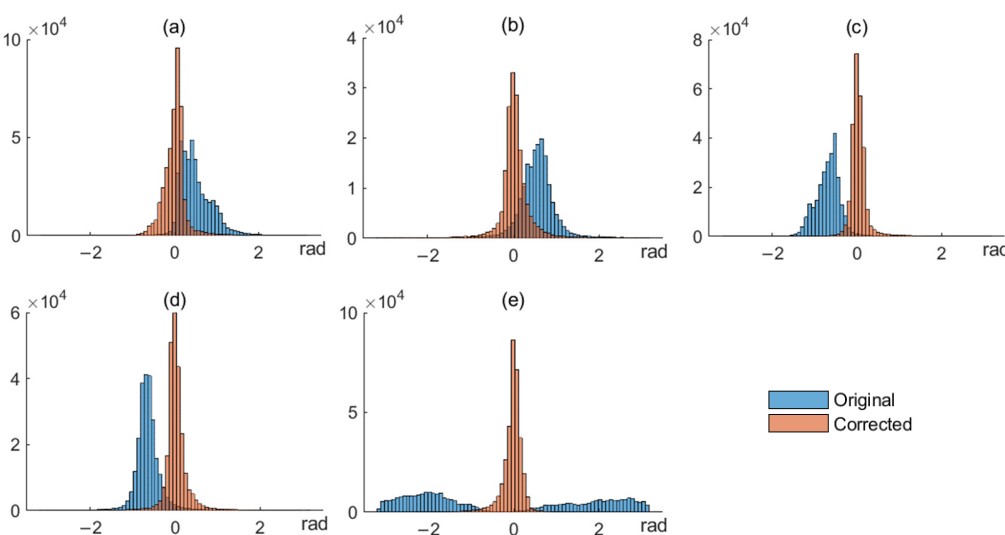

**Figure 5.** Statistical histogram of the wrapped phase before and after the systematic error correction of five typical interference pairs (**a**–**e**) in the experimental area.

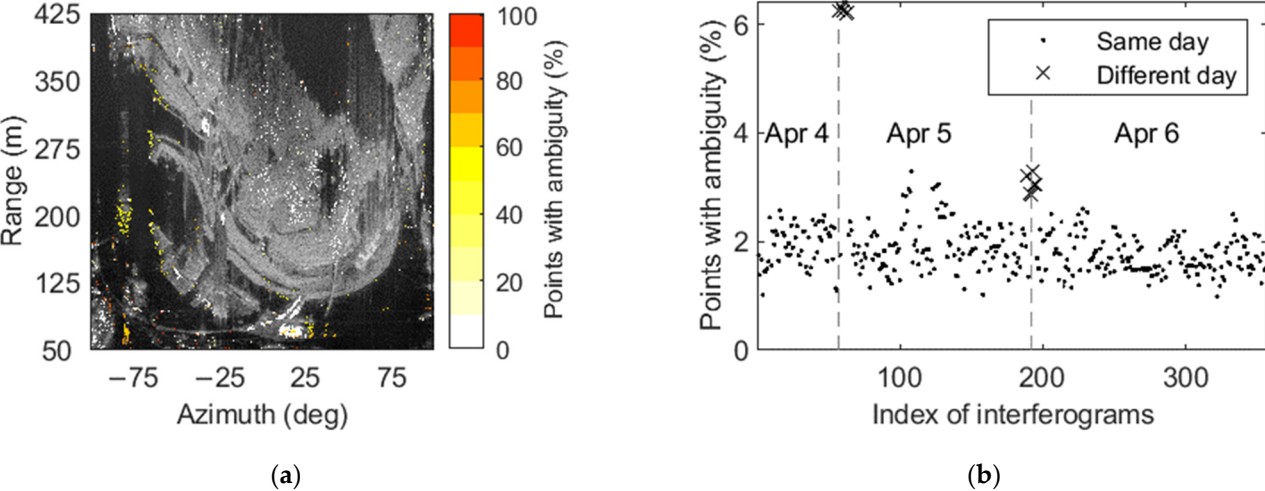

(**a**)                                                            (**b**)

**Figure 6.** Distribution and proportion of points with a phase change before and after unwrapping, i.e., points with ambiguity, and the base image is an intensity gray image. (**a**) Distribution in the spatial domain, the total number is the number of interferograms. (**b**) Distribution of interferograms, the total number is the number of coherent points in an interferogram.

During the next two days, for each SLC acquired, interference was performed with the nearest two SLC scenes to obtain interferograms, and the same method was used to remove the systematic errors. After completing the phase compensation and unwrapping, the cumulative deformation sequence and the average deformation rate were updated using the sequential estimation method in conjunction with the initial cumulative deformation sequence. Figure 8 is the final three-day average deformation rate map, in which three deformation regions, a, b, and c, could be observed. For every point in each deformation region, the cumulative deformation sequence and average deformation rate are shown in Figure 9 and Table 4. The average deformation rate of region a was the fastest, and there was an obvious accelerated deformation process on 4 April, and the deformation rate decreased on 5 April. Regions b and c were relatively stable in the first two days, and their deformation rates increased to 85.49 and 43.81 mm/day, respectively, on 5 April. It is worth noting that the cumulative deformation values were lower in the interval between the two-day observations. There are two possible reasons. On the one hand, the disturbance to the slope was small due to the stop of construction at night. On the other hand, it is more

likely that the deformation amount at night exceeded the half-wavelength of the radar, and the integer number of the deformation phase was lost after phase wrapping.

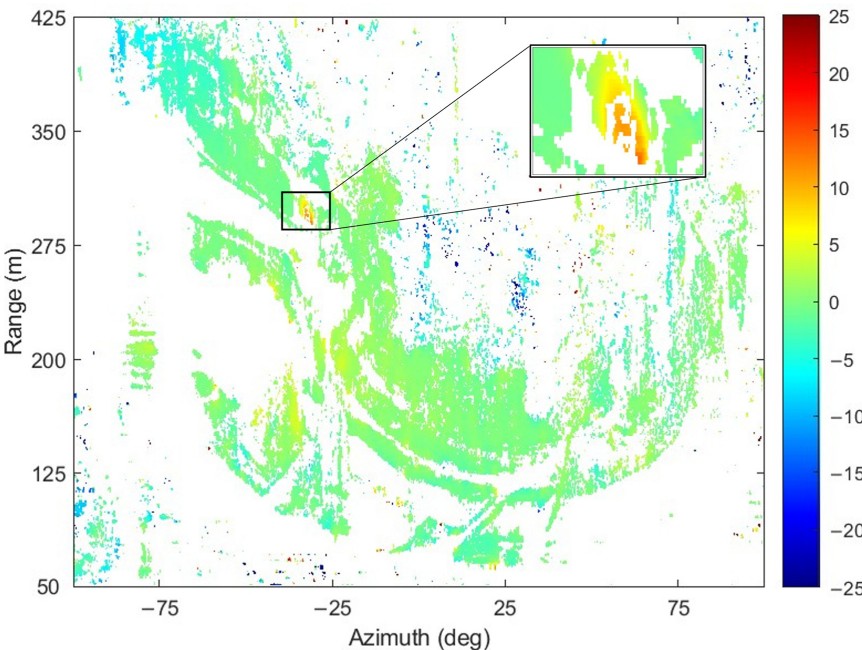

**Figure 7.** Average deformation rate on 3 April 2021.

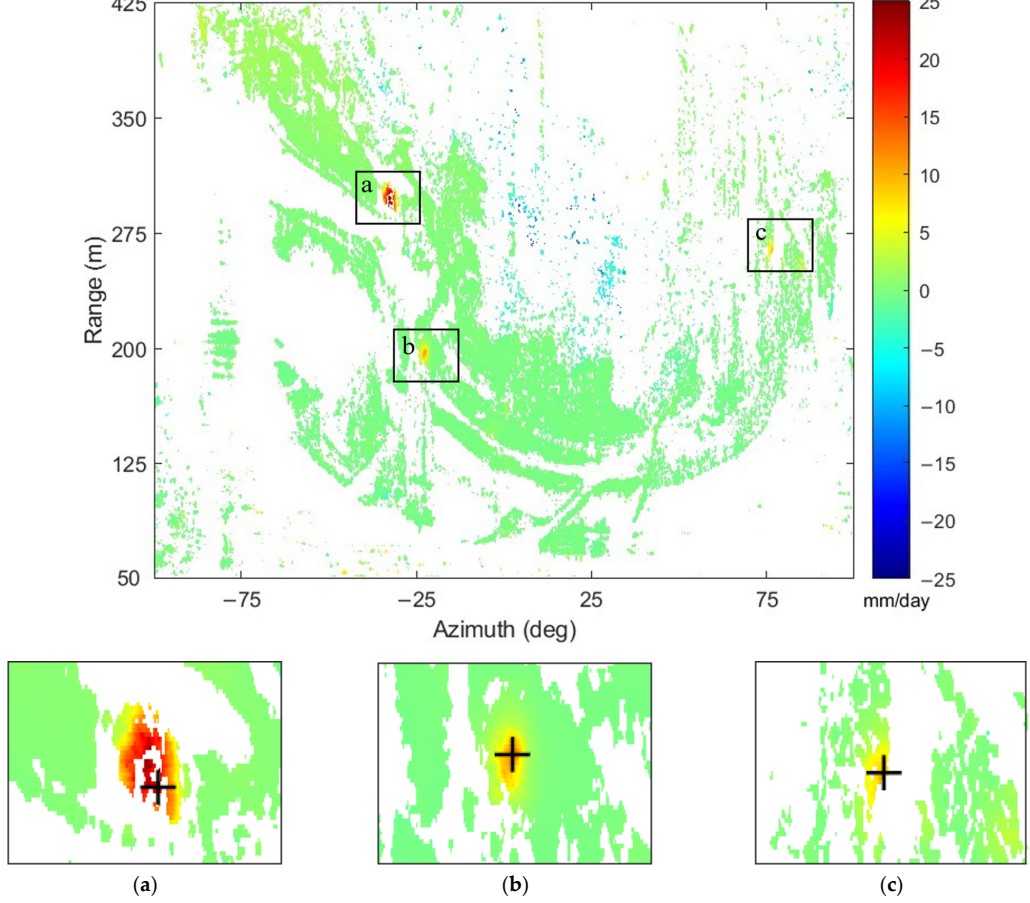

**Figure 8.** The average deformation rate from 3 April to 4 April 2021 obtained with the sequential estimation method and the three significant deformation regions (**a**–**c**).

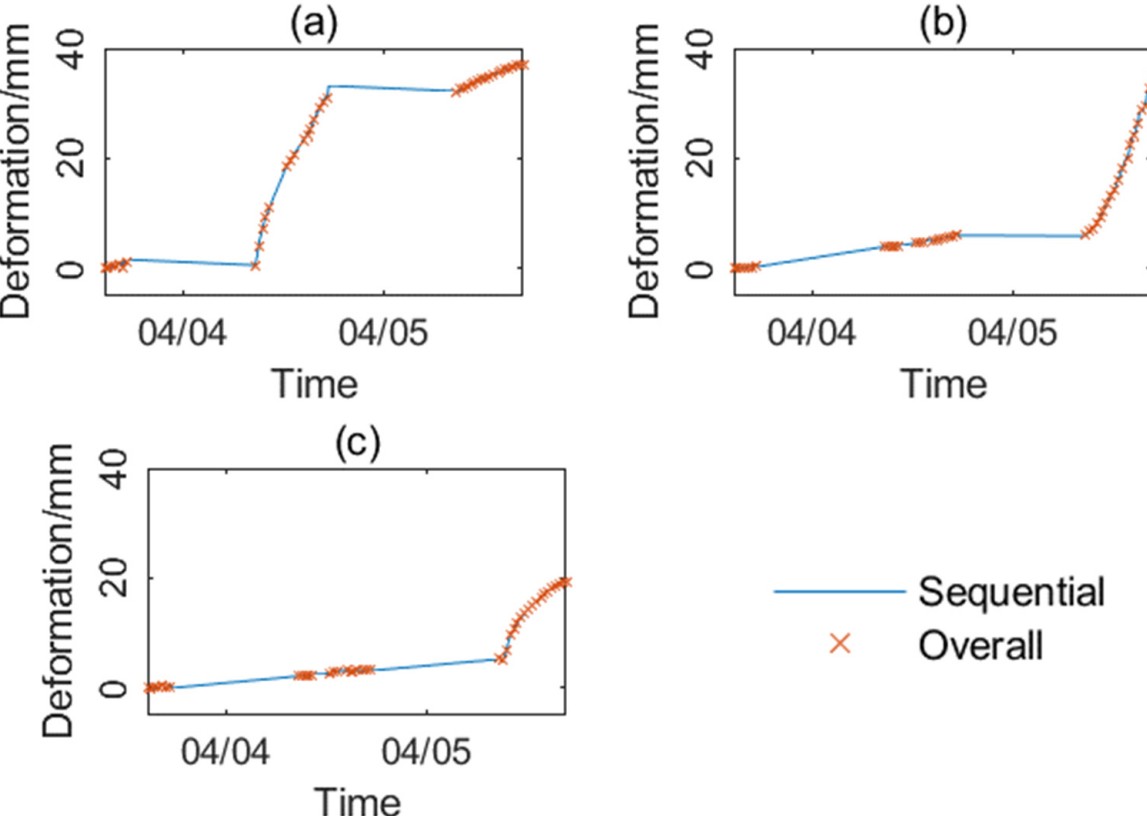

**Figure 9.** Cumulative deformation sequences of regions (**a**–**c**) obtained with sequential estimation and least squares overall estimation.

**Table 4.** The central deformation rate of regions a, b, and c.

| Region | Daily Rate (mm/day) | | | Average Rate(mm/day) |
|---|---|---|---|---|
| | **3 April 2021** | **4 April 2021** | **5 April 2021** | |
| (a) | 10.70 | 78.83 | 15.18 | 18.64 |
| (b) | 3.66 | 5.06 | 85.49 | 8.25 |
| (c) | 2.92 | 3.02 | 43.81 | 8.61 |

Finally, all SLC data were processed with a least squares overall estimation. Figure 9 shows that the two methods had a consistent deformation sequence in the deformation regions a, b, and c. The statistics on the cumulative deformation sequence of all points show that the deviation of the deformation value of 99.35% of the points is less than 0.1 mm, and the mean and standard deviation of the deviation are both better than 0.01 mm. The results obtained by LS overall estimation and the sequential estimation are highly consistent.

Figure 10 recorded the time consumption of the two methods separately, and the programs of the two methods were run independently on a workstation without interference. The data processing started from the 21st period, and the time consumption of the program operation was recorded for each additional period of data. Sequential program time consumption spikes were due to saving data when processing the last period of the day. Nonetheless, the sequential estimation consumed less time than the overall estimation after 21 periods. The average time growth rate of the sequential estimates was approximately 0.018 s/epoch, which is approximately one-tenth of that of the SBAS method (0.182 s/epoch).

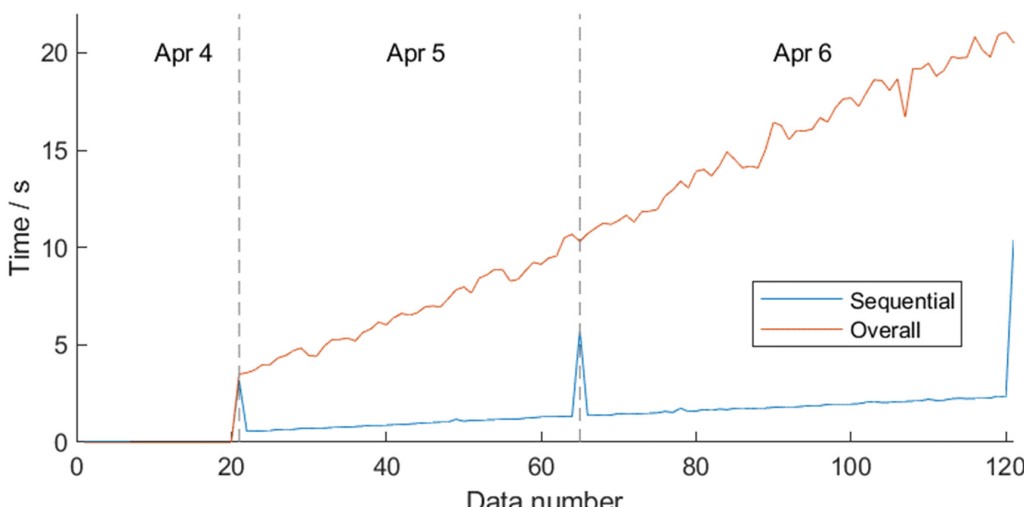

**Figure 10.** Continuous appending data, additional time consumed with sequential estimation, and least squares overall estimation.

## 4. Discussion

The experiment in Zhen'an County, Shangqiu City, Shaanxi Province proved the feasibility of the wrapped phase systematic error correction method and the sequential estimation method in obtaining the slope deformation sequence.

By constructing a two-dimensional polynomial based on a network, the systematic error correction method could estimate and remove the phase error without phase unwrapping. Figure 4 shows that after the systematic error correction, most of the phases tended to be zero, and the phase gradient changes were significantly reduced, which could improve the accuracy and efficiency of phase unwrapping. The comparative analysis of the wrapped and unwrapped phases proved that phase unwrapping is not necessary under specific conditions. First, during repeated observations, the LOS distance change could not exceed a quarter of the radar wavelength, which means that the radar observation interval was short enough or the target deformation was slow. Second, the systematic error led to a wide range of phase ambiguity, and proper error correction was necessary. In this way, the efficiency of data processing could be greatly improved, and the unwrapping error could be avoided. Of course, it is undeniable that in most cases, phase unwrapping could help improve the reliability of the results, especially under complex observation conditions. Furthermore, it should be noted that different systematic error models have different application environments. Generally, the correction accuracy of complex models is higher, but the calculation efficiency is reduced. Therefore, it is necessary to comprehensively evaluate the efficiency and accuracy and use an appropriate model in the rapid monitoring of slopes. At the same time, the current systematic error model is difficult to correct the local atmospheric turbulence caused by extreme weather or complex terrains, which is also the focus and difficulty of phase error research.

The sequential estimation method uses the previous cumulative deformation sequence and a newly added interferogram to obtain a new cumulative deformation sequence according to the principle of least squares. This method has the same deformation accuracy as the least squares overall estimation method and avoids the complexity and repetition of processing the entire data overall. As the amount of data increases, the time consumed by sequential estimation does not grow significantly. This advantage makes it possible to realize real-time monitoring in big data scenarios. The accuracy of sequential estimation is limited by the quality of the observed data; that is, the coherence. However, in the whole observation sequence, the phase quality of the pixel is constantly changing. In order to pursue higher monitoring accuracy, prior weighting (such as correlation) or posterior weighting (such as iterative reweighted least squares or M estimation) could be used. However, these methods inevitably reduce the efficiency of data processing. Therefore,

it is a difficult point, and further research direction is needed to detect and weaken the influence of poor coherence without affecting efficiency.

Due to the limitation of the field conditions, this experiment still has some defects: between the repeated observation intervals, the deformation value exceeded the half wavelength of the radar due to phase wrapping (in Figure 9, after a long interval, the deformation observation change value was close to 0). Additionally, due to the characteristics of GB-InSAR imaging, only the LOS deformation could be observed, which also led to a low observed value. Therefore, it is very important to observe the target continuously and from multiple perspectives in actual slope monitoring.

### 5. Conclusions

In this paper, a near-real-time dynamic GB-InSAR monitoring method for slope stability is proposed. Benefiting from the first correction of the systematic error phase, the phase unwrapping, and the adoption of the sequential estimation strategy, the proposed method could update the deformation sequence of the target with minute-level time resolution and millimeter-level deformation accuracy. The experiment verified that the method could be applied to the deformation monitoring of slopes with steep terrains and risks of instability. With the proposed method, large-area, real-time, high-precision, high-resolution slope deformation data could be used as a preliminary basis for managers and operators to avoid or reduce the damage and losses caused by slope instability.

The method proposed in this paper was developed using MATLAB software, and the demo code and case data can be obtained by contacting the authors.

**Author Contributions:** Conceptualization, Y.S. and H.Y.; Data curation, Y.S.; Formal analysis, Y.S.; Funding acquisition, H.Y.; Investigation, Y.L. and B.Z.; Methodology, Y.S.; Resources, Y.L. and B.Z.; Software, Y.S.; Supervision, H.Y. and J.P.; Validation, H.Y. and J.P.; Visualization, Y.S. and M.S.; Writing—original draft, Y.S.; Writing—review & editing, Y.S. and M.S. All authors have read and agreed to the published version of the manuscript.

**Funding:** This research was funded by the National Natural Science Foundation of China (42174026), the National Key Research and Development Program of China (2021YFE0116800), and the Open Fund of State Key Laboratory of Coal Resources and Safe Mining (Grant No. SKLCRSM20KFA12).

**Data Availability Statement:** The data presented in this study are available upon request from the corresponding author.

**Conflicts of Interest:** The authors declare no conflict of interest.

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
