# Peer review of "A Novel Near-Real-Time GB-InSAR Slope Deformation Monitoring Method"

_remotesensing, doi:10.3390/rs14215585_

Round 1
Reviewer 1 Report
Comments can be found in the attachment.

Reviewer 2 Report
Following the state-of-the-art of InSAR and multitemporal InSAR technology, the authors developed a near-real-time InSAR processing method for the GB-InSAR data by integration the systematic error removal and the seqential estimator. The design of methodology is rational and results are convincing. The further improvement of this manuscript can be in following from my point of view:
1) In the methodological section, a LS solution is applied with the weight matrix. How this weight values are defined, equal weighting or an adaptive weighting is applied? More clarification on this issue is required considering the sensitivity of this factor for the robust estimation.
2) In the validation section, the authors applied the statistical value of the deviation and the standard deviation. In order to enhance the convince of the result for the parameter estimation, the maximum dispersion value as well as the histogram plot of the dispersion values are encourage to be added.
3) In the discussion section, apart from LS method, wether other estimators can be discussed, such as M-estimator that is robust for unknown parameter calucation in case of the occurrence of the aliasing.
4) Finally, more relevant references of GB-InSAR data processing approaches and applications are encouraged to be added.
Round 2
Reviewer 1 Report
To my understanding, although phase unwrapping is saved for error estimation, it's still necessary regarding the whole processing chain from the original radar echoes to the final product. I suggest the authors clarify this point in the discussion section.
One additional suggestion, the reference for the MCF unwrapping method (line 306) should be cited.
Author Response
Thank you again for your valuable comments and recognition of our research. The reply to your question is as follows:
1. Indeed, due to the complexity of the actual observation, phase unwrapping is helpful to improve the reliability of the results. In the discussion, the special condition of not conducting phase unwrapping is added. First, during repeated observation, the LOS distance change cannot exceed a quarter of the radar wavelength, which means that the radar observation interval should be short enough or the target deformation should be slow. Second, the systematic error will lead to a wide range of phase ambiguity, and correct error correction is necessary. In this way, the efficiency of data processing can be greatly improved and the unwrapping error can be avoided.
2. Increase the reference of the MCF unwrapping method.
- Costantini, M. A novel phase unwrapping method based on network programming. IEEE Transactions on Geoscience and Remote Sensing 1998, 36, 813-821, doi:10.1109/36.673674.